# A Synthetic Data Generation Technique for Enhancement of Prediction Accuracy of Electric Vehicles Demand

**DOI:** 10.3390/s23020594

**Published:** 2023-01-04

**Authors:** Subhajit Chatterjee, Yung-Cheol Byun

**Affiliations:** 1Department of Computer Engineering, Jeju National University, Jeju 63243, Republic of Korea; 2Department of Computer Engineering, Major of Electronic Engineering, Jeju National University, Institute of Information Science & Technology, Jeju 63243, Republic of Korea

**Keywords:** deep learning, machine learning, demand prediction, generative adversarial networks, regression, ensemble method, electric vehicles

## Abstract

In terms of electric vehicles (EVs), electric kickboards are crucial elements of smart transportation networks for short-distance travel that is risk-free, economical, and environmentally friendly. Forecasting the daily demand can improve the local service provider’s access to information and help them manage their short-term supply more effectively. This study developed the forecasting model using real-time data and weather information from Jeju Island, South Korea. Cluster analysis under the rental pattern of the electric kickboard is a component of the forecasting processes. We cannot achieve noticeable results at first because of the low amount of training data. We require a lot of data to produce a solid prediction result. For the sake of the subsequent experimental procedure, we created synthetic time-series data using a generative adversarial networks (GAN) approach and combined the synthetic data with the original data. The outcomes have shown how the GAN-based synthetic data generation approach has the potential to enhance prediction accuracy. We employ an ensemble model to improve prediction results that cannot be achieved using a single regressor model. It is a weighted combination of several base regression models to one meta-regressor. To anticipate the daily demand in this study, we create an ensemble model by merging three separate base machine learning algorithms, namely CatBoost, Random Forest (RF), and Extreme Gradient Boosting (XGBoost). The effectiveness of the suggested strategies was assessed using some evaluation indicators. The forecasting outcomes demonstrate that mixing synthetic data with original data improves the robustness of daily demand forecasting and outperforms other models by generating more agreeable values for suggested assessment measures. The outcomes further show that applying ensemble techniques can reasonably increase the forecasting model’s accuracy for daily electric kickboard demand.

## 1. Introduction

A viable alternative that could help achieve the aims of sustainable urban transportation is EVs. Because they do not release polluting gas into the air, EVs are considered eco-friendly. Due to rapid economic and social development, climate change, environmental degradation, and other challenges have drawn continued attention from governments and academics worldwide [1]. Today, many nations see it as a key mission to achieve sustainable transportation to meet future energy requirements. Using EVs significantly improves energy security and lowers greenhouse gas and other pollutant emissions. Imports meet about 97% of South Korea’s primary energy needs due to a lack of native resources.

An evolutionary force in contemporary transportation is shared mobility. Bike sharing, electric kickboards, and electric bicycles are available options. Travelers and students like to avail of a comfortable transportation system. Unfortunately, the drive for private cars has produced some negative environmental side effects, including lengthy delays and congestion. Traffic congestion grows along with population density. In response to this obstacle, shared mobility has come to be recognized as the key paradigm for reducing traffic jams, lowering emissions, and improving accessibility for commuters and students. Making sense of the current data statistically can help manage ride-sharing operations. Finding patterns and drawing lessons from the current data are necessary for demand prediction using various data. Weather information and geographic coordinates may be included in this data for better understanding [2].

To meet the demand for education and travel-centric areas, businesses specializing in electric kickboards have been established in recent years. The high demand penetration may have some significant adverse effects on the new business systems because of the high demand penetration, which includes inconsistent service and fewer number of kickboards in the specific region; most of the kickboards are distributed in a scattered manner. The widespread usage of electric kickboards encourages the implementation of daily demand in a particular industry, which can help the company function safely.

A rapidly developing nation such as South Korea places more emphasis on local market sustainability. Numerous fundamental business presumptions, such as turnover, total revenues, income, capital consumption, etc., are supported by demand prediction. The accuracy of the models can be improved by external factors that can affect demand, such as weather and locational data. A type of time-based data that may be used to visualize various media is time series modeling. However, because conventional statistical methods continue to dominate it, time series analysis has yet to reach its peak. Previous data are investigated, examined, and organized during forecasting to predict the future. However, most extant publications use just one machine-learning model for short-term forecasting techniques. The benefits of mixing many machine learning models, known as ensemble learning, have yet to be thoroughly researched for challenges relating to shared mobility, such as predicting the demand for electric kickboards. This research proposes a brand-new ensemble learning-based method for anticipating daily demand for electric kickboards.

At the beginning of the start-up business module, it is impossible to grow the business significantly, so the data generated by a start-up company should be fewer data compared with existing businesses that have been running their business for 5–10 years in the industry market. Generating synthetic data that are indistinguishable from real data and can be utilized for research and investigation is one approach to resolving the problem of data scarcity. The GAN technique is the most innovative way to deal with this small training data size issue to accomplish considerable prediction results. In some recent work published by the authors, they solved issues such as the data scarcity problem [3,4] and imbalanced data issues solved by a hybrid-GAN approach [5], and the effectiveness of generating synthetic data to improve the accuracy of prediction models has been shown in the articles [6,7]. In this study, we concentrated on creating synthetic time-series data using improved conditional tabular generative adversarial networks (CTGAN), which addressed the following research section.

The summary of the contribution follows:This paper mainly emphasizes synthetic time-series data generation techniques using conditional generative adversarial networks for improving the forecasting accuracy of electric kickboards.The proposed conditional tabular-GAN (CTGAN) model improved the generation of synthetic time-series data that are further combined with original data to enhance the prediction accuracy.An ensemble method was employed to capture uncertainty in forecasting with a single regressor model that is compared on the same dataset with the same preprocessing under the same experimental condition.The importance of the suggested ensemble forecasting model, a comparison with single models and alternative ensemble approaches, and the superiority of the produced forecasting model’s stability are discussed.

The paper’s organization is divided into five sections: The related study, cutting-edge methods for creating various time-series synthetic data, and their implications for prediction are also covered in Section 2. Section 3 provides a summary of the suggested methodology. Analysis and explanation of the experimental demand forecast are in Section 4. We have discussed the present work’s concept and the proposed framework’s predictive performance in Section 5. The conclusion and additional recommendations for future works are described in Section 6.

## 2. Related Works

Forecasting demand for an electric vehicle company is crucial since an electric vehicle-based company planning process is based on the data to be produced. Accurate demand forecasting is needed to fulfill the rising demand. The work completed to forecast daily demand for electric mobility in several fields will be covered in this section along with numerous exemplary studies. Researchers have invested much time and effort in advancing time-series analysis models and forecasting accuracy over the years. To forecast the observed historical data, researchers have put tremendous effort into solving time-series prediction issues. The anticipation of demand for electric mobility has drawn greater attention. They are comparable in terms of data processing, how rental systems are designed to solve problems, and how electric vehicle availability issues vary by location. Therefore, one forecasting technique might be useful for resolving these issues. This section will discuss the associated works with demand forecasting for electric mobility.

The focus is on studies that have predicted the demand for electric vehicles (EVs) at charging stations, parking lots, or traffic situations. The places are pre-labeled and consist of parking lots, charging stations, and traffic situations. Additionally, most publications focus on the impact of prediction passenger demand. Only a few studies, meanwhile, have focused on predicting daily mobility needs. Problems, including a weak billing infrastructure, have also emerged to meet the expanding demand in the EV sector. Effective commercial EV bill demand forecasting enables investment planning and resource allocation for long-term infrastructure bills while ensuring the dependability and stability of short-term network utilities. As a result, we point out a research hole in the daily demand for electric kickboards. Specifically, the technique first determines the important location according to the rent pattern and then foretells the daily requirement for electric kickboards in that location.

Shared electric dockless scooters, often known as e-scooters, provide a convenient mode of transportation for quick trips and are especially well-suited for fostering multimodal interactions [8]. Recent exponential growth was seen in these services, and their rate of adoption was higher than that of other shared modes such as bike and auto sharing [9]. Similar research was also put out [10], employing trip trajectory data from an e-scooter service company and spatial inventory data on the infrastructure to integrate e-scooters into Austin, Texas’s urban infrastructure. They discovered that it uses more than eleven million location points from roughly 80,000 e-scooter rides over the course of a year or 1.4% of all e-scooter trips in the city during the same time period. According to the technique results, a typical e-scooter trip distance is divided between sidewalks (18%), bike lanes (11%), and streets (33%).

A multi-step ride-hailing demand prediction model was developed by Wang et al. [11] using a convolutional neural network (CNN). In this article, the authors show that CNN is 30% faster in the training and prediction process than another deep learning (DL) model, long short-term memory (LSTM). To increase the prediction performance, the author separated the Chinese metropolis Chengdu into smaller zones and afterward added background information, such as weather data. In order to tackle short-term passenger demand forecasting, [12] this study suggests a unique DL method called the fusion convolutional long short-term memory network (FCL-Net). The authors suggested that exogenous, geographical, and temporal dependencies made forecasting models more difficult. In Hangzhou, China, data were gathered for the experiment’s short-term passenger demand forecasting under an on-demand ride service platform.

The hybrid model was proposed [13] using the autoregressive integrated moving average (ARIMA) to obtain higher forecasting accuracy. In the article [14], the hybrid model was superior to the individual ML models in terms of error metrics for predicting short-term traffic flow. The prediction of traffic states is a significant issue with significant consequences for contemporary traffic management. Using a neural network, a local traffic status estimate and prediction method are provided [15]. To explain endogenous links between variables in the taxi market under the two-sided market equilibrium, a model was created [16]. The waiting time of passengers and the cost of a cab impact accurate passenger demand on the demand side. The authors [17] propose using real-time taxicab data to estimate passenger demand and taxicab supply across urban regions. The authors’ [18] primary goal is to show that maximum predictability can be attained by utilizing appropriate measures when choosing prediction algorithms. They used two sets of real-world data from Uber and yellow cab rides in New York City. In the experiment, they first measured the demand uncertainty at the building block level using entropy and the temporal correlation of human mobility. Second, the outcomes of the three prediction algorithms were contrasted. Regarding prediction accuracy and calculation time comparison, the Markov predictor performs well.

On-demand transportation service platforms can link waiting for passengers with available registered vehicles efficiently because of the quickly developing internet technology. The demand service market is dependent on customers and their actual riding experiences in the various modes, just like the electric mobility service. The waiting time and fare of the customers had an impact on accurate passenger demand on the demand side. On the other hand, the predicted seeking time on the supply side altered the rider’s behavior or where to park the electric kickboard and reduced the rent. Xu et al. [19] proposed a taxi demand service based on a neural network for the prediction demand of taxis in a specific area. Amini et al. [20] introduce electric vehicle charging demand forecasting depending on historical driving data. By fusing CNN and RNN, numerous spatiotemporal prediction methods were used to describe the predicted trip demand’s temporal and spatial correlation [21,22,23].

A model was put forth to increase the effectiveness of the taxi service or bike-sharing systems by forecasting the demand for pick-up or drop-off the next time by merging the spatiotemporal neural network with the LSTM model [24]. Using a neural network-based approach enables short-term multi-zone passenger demand prediction. The Didi Chuxing, Chengdu, China, car-calling demand data and the New York City taxi demand data were the sources of the information used for the experiment [25].

A variety of ensemble approaches were used in a recent study on demand prediction to anticipate the charging time for EVs [26]. Eddine et al. [27] suggested a new deep learning method for charging demand predicting for EVs that achieved good prediction results by experimenting with two datasets. The location chosen for the study was based on how frequently cars were utilized. A complete charging demand prediction model is established [28], taking into account the charging factors of electric vehicles, and the travel behavior of electric vehicles under various spatiotemporal distributions in the Second Ring Road area of Chengdu is simulated by Monte Carlo. The study is divided into functional areas based on the urban point-of-interest data crawled by Python. The simulation results demonstrate the effectiveness of the suggested strategy for predicting charging demand in various contexts and circumstances.

Data scarcity is considered one of the most significant challenging issues in machine learning. Developing a demand prediction model needs a good chunk of data for training a machine learning model. Real-time data always have a drawback, which is small data. Little research has been developed for generating synthetic data for improving prediction performance. Predicting the consequences of energy production and consumption becomes increasingly important with the growing share of renewable sources in electricity generation. Due to privacy issues, the historical data that the models are based on is constrained. GAN has been used to create synthetic new data and is a promising approach for producing realistic synthetic time-series data. The authors [29] created synthetic data for predicting wind power series using the GAN model. Solutions include generating realistic time series with the aid of generative models, augmenting real-world financial data taken from the US stock market, and demonstrating the value of daily market prediction findings [30]. In this study [31], pairing the actual data with created synthetic data using the GAN model could lower the prediction error of electricity consumption. To close the gap in the production of synthetic data in a data center that will impact forecasting in the energy industry, time-series data augmentation based on GANs is used [32].

## 3. Methodology

A brief explanation of the information and methodology is provided in this section. This study aims to improve forecasting precision for the daily demand for electric kickboards among local service providers. In this study, we create a new ensemble forecasting model that combines the advantages of individual forecasting models to produce forecasts with improved accuracy and stability. This study suggests a new method for predicting daily demand for an electric kickboard startup business in the province of Jeju Island. We have built a system for developing synthetic data to address the data scarcity issue, which could be helpful in our research. The dataset for real-world electric kickboards is used to implement the framework. Initially, we faced problems achieving noticeable results because we needed more training data. We require a lot of data to produce a solid prediction result. We employed the GAN-based approach for generating synthetic time-series data. The original data have been preprocessed before being sent to the proposed GAN model, which will create synthetic data to improve the prediction model’s performance. We have used conditional synthetic data generation for those rent stations where the actual rent is lower on the side. That model will generate synthetic data only for those conditional rent stations. This study also examines the capability of predictive regression ensemble models by contrasting the ensembles and considering the single reference models to estimate the demand. Figure 1 shows the proposed framework’s overall layout. We considered current information from the regional business in South Korea’s Jeju province.

The subsequent step is preprocessing, when we prepare the data for our experiment by combining the data from many sources into a single final data set. Then, we determine whether or not there is a null value. The prediction’s importance and utility were considered when choosing the data features. We used a customized version of the k-means clustering module. The data distribution has led to the selection of four clusters. We first trained the model using the original data to obtain the prediction result. The GAN model was then trained to produce synthetic data using the input from the original data. Then, we blended the original and synthetic data using the original date distribution. Before training a final model with these chosen settings on the entire training set, cross-validation (CV) finds the settings that produce the least amount of error. This function uses cross-validation to train and assess the performance of each estimator in the model library. Models were developed and evaluated using 10-fold cross-validation. All of the estimators in the model library are trained and evaluated using cross-validation using the compare_model function in PyCaret. This function produces a scoring grid with typical cross-validated scores as its result. We create an ensemble regressor technique that uses machine learning models with the best hyperparameters as the base model and evaluation metrics to assess the accuracy of the prediction made using the suggested framework.

### 3.1. Data Analysis

The experiment’s data collection is briefly described in this section. We used data from a local electric kickboard service provider on Jeju Island. The dataset collected the rent details from 16 April 2019 to 11 June 2021, including the spatial and temporal data events. Additionally, meteorological data were imported via the Korea Meteorological Administration. We considered external factors that affect how kickboard services are used, such as average daily temperature, rainfall, and the separation of weekends and weekdays. The dataset contains original real-time electric mobility data (CSV), temporal data (CSV), and spatial data (CSV) obtained by the local company. Several processes were involved in preprocessing the complete data. To ensure data consistency, we have also eliminated the rows with missing data, decreased inconsistent values, and removed duplicate data. The data are meticulously examined before the experiment, and the date format conversion is completed before using the final data. Saturday and Sunday are regarded as holidays in the later preprocessing on the day when weekdays and weekends are filtered. Using the preprocessed data, we use the k-means clustering algorithm to identify several clusters on Jeju Island. The features were chosen based on how useful and significant they were for forecasting daily demand. After feature selection, a total of 13 features were utilized. Figure 2 shows the rental locations of electric kickboards, where the *x*-axis depicts the latitude as xpos, and the *y*-axis indicates the longitude as ypos. The parameters or coordinates latitude and longitude are used to pinpoint the position of any site.

The regional data were divided into four clusters according to the rental location data distribution. First, we divided Jeju Island into Jeju City and the Seogwipo City region. Jeju city is further divided into three clusters. To cluster the data in accordance with the rent pattern, we used k-means clustering as shown in Figure 3, four clusters with the names sector 0 through sector 3 have been chosen as a result of the data distribution, where xpos represent latitude over the *x*-axis and ypos represent longitude over the *y*-axis. Other selected features are temperature, insulation, humidity, day, month, year, weekend, rain, and holiday. Figure 3 displays the variation in dot size based on the total number of rentals at each Jeju Island rental station over the data collection. Figure 4 depicts sector-wise rent demand by the passenger. Electric kickboard rentals are most popular in sector 1, while the demand for kickboards is lowest in sector 0. The *y*-axis lists the total rent for each sector, and the *x*-axis lists the various sector numbers.

### 3.2. Generative Adversarial Networks

In 2014, Goodfellow et al. [33] created generative adversarial networks, which is a kind of neural network utilized in unsupervised learning methods. Image, tabular, and time-series data fields have been expanded using the GAN technique. In this study, we used GAN to create synthetic time-series data and enhance the performance of the regression model. GANs are composed of two neural network models that are in competition with one another for superior prediction accuracy. The discriminator and generator are the names of the two neural networks. A deconvolutional neural network serves as the discriminator, and a convolutional neural network serves as the generator. The generator aims to produce outputs that may be mistaken for real data. The discriminator aims to distinguish between the real and artificial data it receives.

Equation (Equation 1) illustrates a mathematical function regarding the cross-entropy minmax game between these two networks.
(1)$$\min_{G}\;\max_{D}\;R\left(D,G\right)=\underset{︸}{E_{x∼P_{data}\left(x\right)}\left[logD\left(x\right)\right]}+\underset{︸}{E_{rz∼P_{rz}\left(rz\right)}\left[log\left(1-D\left(G\left(rz\right)\right)\right)\right]}$$

Data are generated by generator *G* using inputs of real data x and random noise variable rz. The produced data returned by this process, G(rz), should match the actual data distribution. D(x) represents the probability that discriminator D would identify and classify *x* as actual data, where *E* denotes the expectation. D(G(rz)) is the probability that *D* will classify the data as having been generated by *G*. The objectives of the generator and discriminator are in opposition. By making D(G(rz)) approach 0, which shows that D can precisely identify fake data from real data, *D* seeks to enhance this error while *G* seeks to reduce it. Figure 5 depicts the GAN’s basic architecture.

### 3.3. Conditional Generative Adversarial Networks

When in terms of time-series tabular data, the algorithms can only produce data with the same distribution as the real data if there are concurrently discrete and continuous variables in the actual data. In 2019, Xu et al. [34] suggested the CTGAN model, an upgrade above TGAN, to address this issue. The goals of CTGAN are very similar to those of TGAN [35]. CTGAN is different because it is more ambitious. It seeks to preserve the joint distribution of all columns rather than merely the correlation between any two columns in the synthetic data. Learning the true distribution of data is one of the conditional generator’s most crucial jobs. Given a constraint on a certain discrete value, the generator’s goal is to change the input to the original distribution as Equation (Equation 2).
(2)$$P\left(\mathrm{row}\right)=\sum_{k\in C_{i}^{*}}P_{G}\left(\mathrm{row}\;|C_{i}=k\right)P\left(C_{i}=k\right)$$
where *i* stands for a particular discrete attribute, while *k* represents values from the discrete column $C_{i}$.

Synthetic data generated using the improved CTGAN model’s configuration are shown in Table 1. The residuals determine the size of the output samples, in this case, $generator_{d}im$. These residuals will increase when more numbers are added to the list: one for each number. This has the same effect as deepening the generator. The size of the output samples for each linear discriminator layer is also represented by $discriminator_{d}im$. As for the quantity of training iterations the model will experience, we have chosen 600 epochs for more appropriate training. A larger epoch number will generally lead to longer training times but better synthetic data.

### 3.4. Ensemble Learning Model

With the ensemble method, learning tasks are accomplished by assembling and combining a number of weak learners. The primary learners are trained using the two well-known techniques, bagging and boosting. Bagging reduces variance, which enhances the generalization of the model. To increase the model’s accuracy, boosting uses a variety of machine learning (ML) approaches to turn a number of weak learners into strong learners. The ensemble approach is chosen because it is adaptable and has high computational throughput for large-scale dataset prediction operations. Due to their benefits, the ensemble approach has drawn a lot of interest in recent years. Voting regressors are ensemble models that combine many base models’ predictions into a single, final forecast. The ensemble’s base estimators average predicted target value serves as the final forecast. Using a mean and weighted average is the most straightforward way to group regression ensembles. To increase the predicted accuracy of the chosen models, the regression ensemble models build a collection of models. The predictive regression ensemble models have the ability to estimate demand by contrasting the individual models and considering the meta-model.

Using the ensemble approach, a model is created to predict the daily demand for electric kickboards. This strategy’s fundamental tenet is to combine basic models to produce a composite prediction model. To create the ensemble model [36], we used the CatBoost, RF, and XGBoost machine learning algorithms to predict the daily demand for electric kickboards.

#### 3.4.1. CatBoost

Prokhorenkova et al. [37] developed the gradient boosting decision tree (GBDT) technique into categorical boosting (CatBoost). It utilizes category features efficiently and with the least amount of information loss. It is distinct from other gradient boosting techniques and starts using ordered boosting, which is a successfully adapted method. This method can handle category features and does well with small datasets. This procedure, frequently carried out during preprocessing, mainly entails replacing the original category features with numerical values.

It has recently been used in various industries, including finance, and in different data sources, including time-series data. Although CatBoost may not be the greatest learner for homogeneous data, it is an excellent answer for heterogeneous data issues. In CatBoost, a new binary feature is added in place of the original variable for each category. It chooses the tree structure using random permutations to determine leaf values and prevent overfitting by conventional gradient-boosting methods.

#### 3.4.2. Random Forest

Using random forests (RF), ref. [38] is a common ensemble method. It is frequently applied to classification and regression issues. During training, the RF algorithm aggregates decision trees and produces the mean forecast for each tree (regression). The model generates a massive forest of arbitrary, uncorrelated decision trees to find the best possible solution in RF. By only choosing a portion of the feature space at each split, RF makes an effort to overcome the correlation problem. The RF method functions effectively on enormous data sets for various applications and has the highest accuracy of the known algorithms. Without destroying any of them, it can handle a huge variety of input variables. A straight inside estimation of the generalization error is produced as the forest expands. Numerous separate decision trees are used in the RF technique, each formed from a different subset (bootstrap sample) of the training data. In the tree-building process, the best split for each node is chosen by randomly choosing candidate variables.

#### 3.4.3. Extreme Gradient Boosting

The “boosting” idea, which combines the prediction of numerous weak learners with additive training methods to create a strong learner, is the foundation of the extreme gradient boosting (XGBoost) technique, which was put forth by Chen et al. [39] in 2016. This freshly created algorithm has found widespread use in many different industries. XGBoost uses a regularized technique, formalization, to prevent overfitting and achieve better performance. The integrated framework employs a random sampling strategy to lower variance and improve the final model’s predictive capability.

## 4. Results and Discussion

Algorithms for ML were developed utilizing the google collaborative framework. To read the data and implement the models, several ML packages, including NumPy, pandas, sklearn [40], seaborn, etc., were imported.

### 4.1. Accuracy Measurement Metrics

The discrepancies between the actual observations in the test set and the predictions made by the final model are used to assess a model’s accuracy. $R^{2}$ and MAPE are the evaluation measures applied in this work. Every regression study uses $R^{2}$ as a critical performance parameter to quantify model prediction. In contrast, a model with a zero $R^{2}$ generates poor forecasts, and one with a value close to one $R^{2}$ yields good predictions. A simple average of absolute percentage errors is known as MAPE. Equations (Equation 3) and (Equation 4) are written in the section below.
(3)$$R^{2}=1-\frac{\sum_{k=1}^{m}{\left(y_{k}-{\hat{y}}_{k}\right)}^{2}}{\sum_{k=1}^{m}{\left(y_{k}-{\bar{y}}_{k}\right)}^{2}}$$
(4)$$MAPE=\frac{1}{m}\sum_{k=1}^{m}\left|\frac{\left(y_{k}-{\hat{y}}_{k}\right)}{y_{k}}\right|\cdot 100\%$$
where $y_{k}$ is the actual value of the $k^{th}$ sample case, ${\hat{y}}_{k}$ is the predicated value of $k^{th}$ sample case, ${\bar{y}}_{k}$ is the average value, and *m* is the sample size.

The objective of the current study is to evaluate the efficacy of the suggested model in estimating the daily demand for electric kickboards using real-time data. The dataset was divided into training (90%) and testing (10%) for the original data and combined synthetic and original data. The parameter settings that directly impact the model’s performance are essential. The choice of appropriate parameter tuning may considerably affect the predictive model’s performance. Using scikit-learn and a grid search method, the ideal parameters that lead to the most incredible performance for the methods have been found. The ideal key parameter for the methods applied in this investigation is shown in Table 2.

### 4.2. Model Performance Comparison

A thorough comparative analysis was completed to compare the proposed model’s applicability and effectiveness for predicting daily demand for electric kickboards with other ML models. Various evaluation metrics, such as MAPE and $R^{2}$, are used to evaluate the performance of the suggested model. A higher $R^{2}$ score and a low MAPE score can be used to describe the model’s performance. Table 3 compares the proposed model for predicting daily demand for electric kickboards with other single ML models. The proposed model also incorporates elements from other ML models. In comparison to previous individual-based models, the suggested ensemble model fits the data the best and has the lowest errors in terms of MAPE value. It also has the highest $R^{2}$ value of any model. As shown in Table 3, the findings indicate that the suggested model outperforms basic models such as CatBoost, RF, XGBoost, ET, and LGBM.

Table 4 lists the outcomes based on two evaluation indicators ($R^{2}$ and MAPE). As seen in Table 4, the ensemble models with the highest $R^{2}$ and lowest MAPE are compared. For the test data set, distinct ensemble models were created for this investigation. The suggested method outperformed all ensemble approaches in a detailed comparison of findings with the lowest MAPE (21.22) and highest $R^{2}$ (0.79) values for combined data. The proposed model has the highest $R^{2}$ (0.64) and lowest MAPE (24.51) values compared to the original data. The proposed model is the best algorithm for predicting daily demand for electric kickboards in all the situations taken into account, according to the thorough analysis based on evaluation metrics ($R^{2}$ and MAPE).

Figure 6 shows a correlation between the actual and predicted demand. Using the test set illustrates the link between actual and predicted demand. The date is shown by the *x*-axis, while the *y*-axis represents the rent count value. The visual depiction of Figure 6 illustrates how the original data forecast result fell short of expectations. On the other hand, the prediction accuracy increased when we used the combined data. The outcome illustrates the suggested model’s improved prediction ability.

## 5. Discussion

At the beginning of every business, a reliable business model must be built to meet everyday demand while offering excellent customer service. Businesses need help to meet demand at the right location, and machine learning can aid by precisely forecasting the demand for electric kickboards. This study provided an electric kickboard demand forecasting model that took into account actual rent data, weather data, and condition data processing together with real-world weather data. These process data also comprise a cluster analysis to categorize clusters according to rental locations. We need a significant amount of data to train a machine-learning model. However, this study initially needs to improve in terms of small data size because a new business is starting. In order to improve prediction accuracy, this study suggests using a conditional tabular-GAN (CTGAN) model to create synthetic time-series data and further blend it with original data. We looked into whether using synthetic data improved prediction accuracy.

Figure 6 illustrates how models can learn from data behavior when they can make predictions using combined data that are consistent with the measured values. The regression ensemble methodology is dependable in reaching predetermined predictions because the increased performance gained during the training stages with combined data is maintained during the test phases. Additionally, the proposed demand forecasting model would make it possible for electric mobility service providers to plan future production and operation. The proposed demand prediction model may also help formulate investment and operation strategies for flexible electric mobility infrastructures based on the demand for electric vehicles.

## 6. Conclusions

The main goal of this study was to predict the daily demand for electric kickboards based on the historical pattern of rental payments using real-world data from a local Jeju Island electric kickboard service provider. The collected data were merged into one single final data for the start of the development. Later, we employed the GAN method for the data scarcity issue that will generate synthetic data as learned from the original data distribution. Integrating the produced synthetic data with the actual data will boost the prediction accuracy of the daily demand for kickboards. The objective of this study was to forecast the daily demand for kickboards using an ensemble model technique. There were three ML algorithms used: RF, CatBoost, and XGBoost. The suggested ensemble method’s prediction performance was evaluated using a variety of measures. The proposed method’s prediction performance was assessed based on the test dataset. The results demonstrated that the proposed ensemble model outperformed other chosen ML models. When we integrated synthetic data with the original data, the performance of the suggested ensemble approach improved, according to a thorough comparison with the original and combined data. The original data produced the highest MAPE value (24.51) and the lowest value of $R^{2}$ (0.64). The model also produced the highest $R^{2}$ (0.79) and lowest MAPE (21.22) values for combined synthetic and original data.

Future research can examine a few potential limitations of this work. To generate synthetic time-series data, we can utilize a more advanced GAN model, various optimization methods, and improved hyperparameter settings for the different ML models. Future studies that consider these variables may offer fascinating insights into predicting daily demand. Furthermore, future studies may use other sophisticated machine and deep learning models.

## Figures and Tables

**Figure 1 sensors-23-00594-f001:**
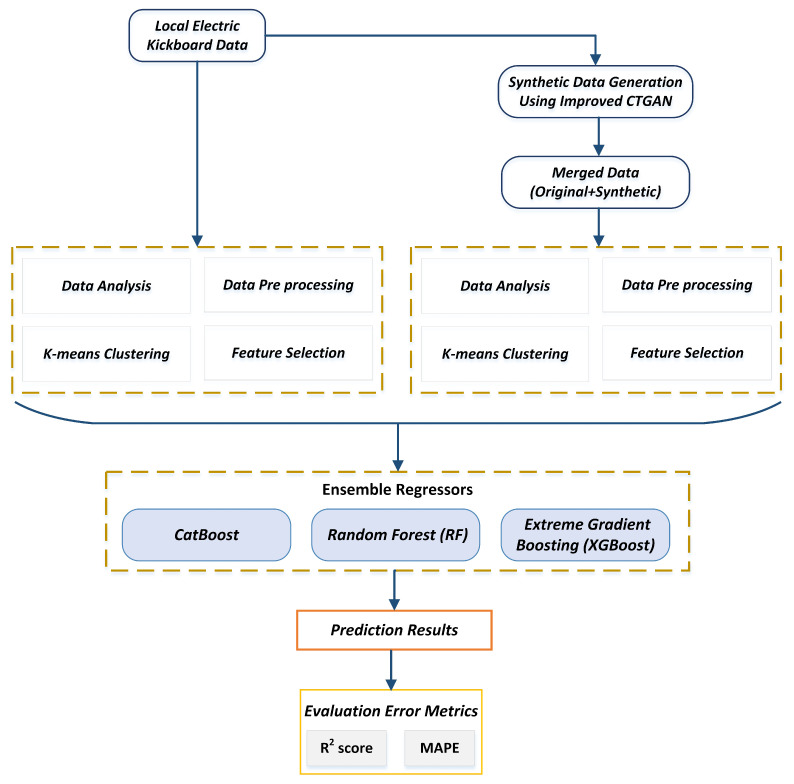
Proposed methodology overview of the study.

**Figure 2 sensors-23-00594-f002:**
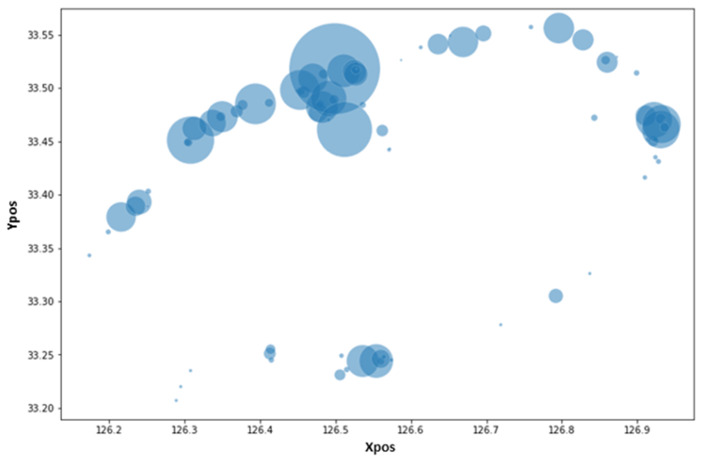
Distribution of rental locations between Jeju City and Seogwipo City.

**Figure 3 sensors-23-00594-f003:**
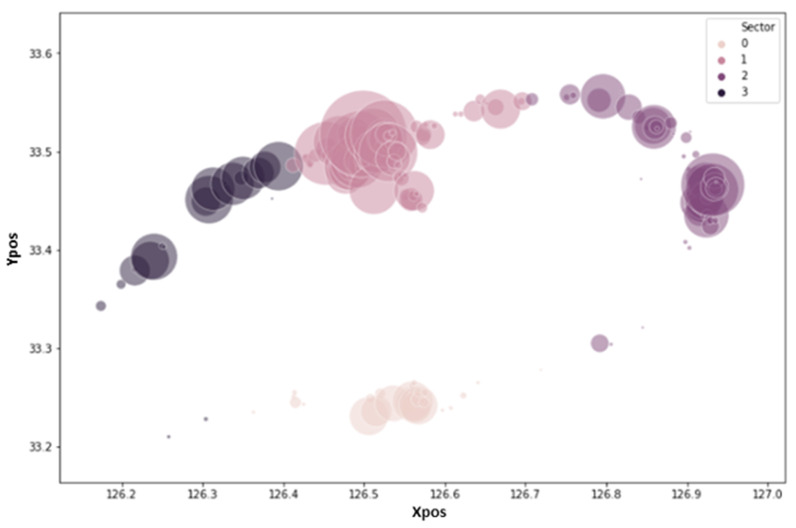
Visualization graph for sectors divided using the clustering method.

**Figure 4 sensors-23-00594-f004:**
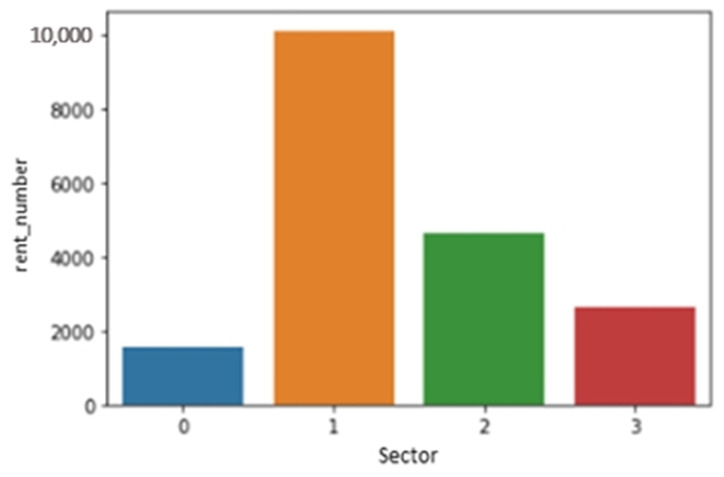
Sector wise rent demand graph.

**Figure 5 sensors-23-00594-f005:**
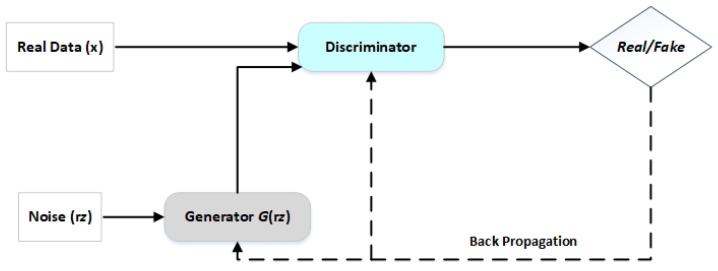
GAN architecture.

**Figure 6 sensors-23-00594-f006:**
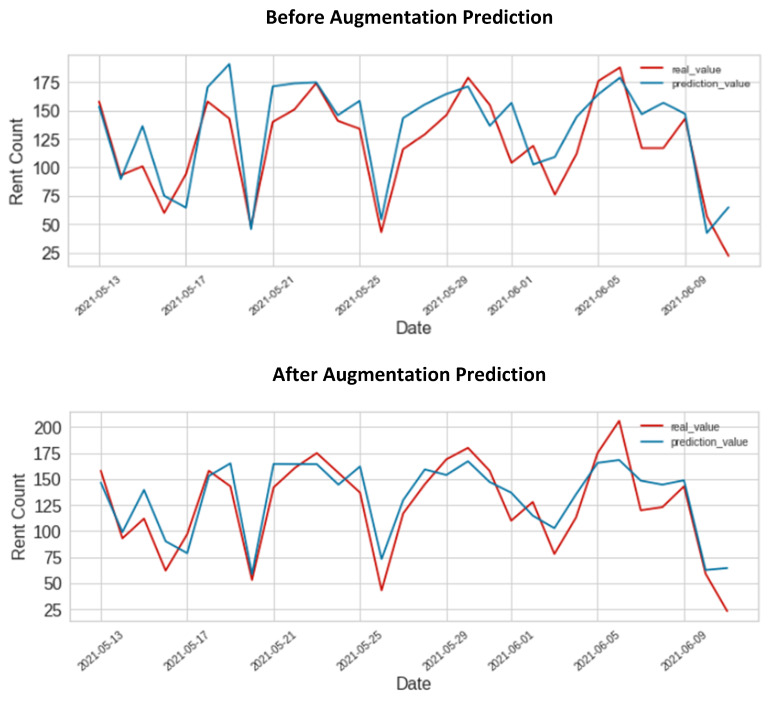
Plot of the proposed regression model prediction for test data compared with the original data and combined synthetic with original data.

**Table 1 sensors-23-00594-t001:** Synthetic data generated using the improved CTGAN model’s configuration are shown below.

Model	Generator_dim	Discriminator_dim	Batch_size	Epochs	Generator_lr	Discriminator_lr	Optimizer
Improved CTGAN	[1024, 1024]	[1024, 1024]	600	600	0.0001	0.00033	Adam

**Table 2 sensors-23-00594-t002:** Parameter settings for model tuning.

Models	k-Fold	Max-Depth	Learning-Rate	n-Estimators
Catboost	10	03	0.3	300
RF	10	12	0.2	200
XGBoost	10	10	0.1	100

**Table 3 sensors-23-00594-t003:** Performance comparison with different single machine learning models.

Models	Original Data		Combined Data (Original + Synthetic)	
	$R^{2}$	MAPE	$R^{2}$	MAPE
RF	0.58	29.21	0.67	25.61
CatBoost	0.57	30.73	0.63	27.11
XGBoost	0.58	29.73	0.69	25.15
Extra Tree (ET)	0.51	31.38	0.49	32.14
LGBM	0.44	39.89	0.46	34.12
Proposed Model	0.64	24.51	0.79	21.22

**Table 4 sensors-23-00594-t004:** Performance comparison with different ensemble approaches.

Ensemble Models	Original Data		Combined Data (Original + Synthetic)	
	$R^{2}$	MAPE	$R^{2}$	MAPE
RF + CatBoost + LightGBM	0.62	26.21	0.72	22.93
Stacking (KNN + DT + RF)	0.59	28.73	0.70	24.11
Stacking (KNN + DT + RF + XGBoost)	0.61	28.25	0.69	25.15
Extra Tree + XGBoost + RF	0.62	26.73	0.71	23.14
Proposed Model	0.64	24.51	0.79	21.22

## Data Availability

Not applicable.

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
