# Peer review of "A Synthetic Data Generation Technique for Enhancement of Prediction Accuracy of Electric Vehicles Demand"

_sensors, 2023, doi:10.3390/s23020594_

Round 1

Reviewer 1 Report

Chatterjee and Byun proposed to use GAN and ensemble learning to predict daily demand of electric kickboards. They tried different learning algorithms such as RF, catBoost, XGBoost, extra Tree and LightGBM as weak learning algorithms, and tested different combinations of them for ensemble prediction.  However, the methodology and evaluation are not clear enough to evaluate the work.

1. There is a flowchart about the data processing (Figure 1). But it is not clear how k-means clustering and feature selection work together for next step. Meanwhile, there is no description about the size of the data for each step.

2. Cross-validation is used to tune hyperparamerters. But It is unknown how to evaluate the performance independently. 5-fold cross-validation, or 10-fold cross-validation? How many times?

3. The reported performance is mainly about single learning algorithms and ensemble learning algorithms. No test/comparison/discussion against existing methods.

4. It is hard to understand the clear picture of related works in “Related works”

5. The manuscript needs a significant revision, since it includes a lot of typos.

Reviewer 2 Report

The title of the research paper: A Synthetic Data Generation Technique for Enhancement of Prediction Accuracy of Electric Vehicles Demand.

The authors presented the required data collection method and generated the synthetic data by using GAN.

They have selected the required features from the dataset and do the preprocessing data and feature extraction from the dataset.

The authors used the ensemble model by combining CatBoost, Random Forest and XGBoost machine learning algorithms.

The results are supported with their claim and explained neatly.

The authors need to do English spell check and grammer checking for entire article.

In figure, the X and Y axis units are not clear. Need to improve. Add explanation after the every figure or inference from the pictorial representation of the results.

Reviewer 3 Report

In this paper, the authors create an ensemble model by combining three machine learning algorithms, such as CatBoost, Random Forest (RF), and Extreme Gradient Boosting (XGBoost), in order to predict daily electric kickboard demand in Jeju Island, South Korea, where the authors made synthetic time-series data using a generative adversarial networks (GAN) approach and combined the synthetic data with the original data. Overall, the authors have made a good attempt. However, due to the lack of comparison with conventional techniques, the effectiveness of the proposed technique is not clear. Besides, due to thin theoretical explanation about the proposed technique, this seems a simulation report. The reviewer’s other comments are as follows:

1.      The quotation of previous articles is rough. For example, “Given the underlying foundation, the GAN technique might be used to accomplish this [3–7]”, “Electric vehicle charging demand prediction proposed by several proposed works [19–23]”, etc. These citations are meaningless. The authors must quote articles properly.

2.      The problem definition of this work is not clear. In Sect. 2, the drawbacks of each conventional technique should be described clearly. The authors should emphasize the difference with other methods to clarify the position of this work further.

3.      The authors should use clear images. Some of the figures are blurry.

4.      Eq. (1) has editing problems.

5.      Overall, the authors have made a good attempt. However, the authors' proposed method does not adequately describe their data. The results are not supported by any theoretical/mathematical reasons. Readers will fail to understand the scientific contribution of this research. The authors should justify the effectiveness of the proposed technique theoretically.

6.      Which articles did you compare with the proposed technique in Table 4? Indicate the reference number in sentences. Besides, the authors must cite the compared articles in References.

7.      The effectiveness of this work is not clear. Through simulations/experiments, the authors must justify the effectiveness of the proposed method by comparing with the other latest methods. Several articles are discussed in the research survey. However, no comparison is shown with these techniques. Frankly speaking, the research survey and References are meaningless. Please show comparison data.

8.      The results of this research are not clear in Conclusions. Furthermore, the benefits of the proposed method are not supported by theory. So, I fail to understand the scientific contribution of this research.

Round 2

Reviewer 2 Report

The authors carefully read the article and include / carried the suggested corrections / improvements in the article.

Reviewer 3 Report

In this paper, the authors create an ensemble model by combining three machine learning algorithms, such as CatBoost, Random Forest (RF), and Extreme Gradient Boosting (XGBoost), in order to predict daily electric kickboard demand in Jeju Island, South Korea, where the authors made synthetic time-series data using a generative adversarial networks (GAN) approach and combined the synthetic data with the original data. In the revised version, most of the reviewer’s requests were met by the authors. The reviewer would like to pay tribute to the authors’ great work. This is well written and organized paper. It is scientifically sound and contains sufficient interest to merit publication, I think.
